# Delays in Treatment Initiation and Treatment Outcomes in Patients with Tuberculosis in the Kyrgyz Republic: Are There Differences between Migrants and Non-Migrants?

**DOI:** 10.3390/tropicalmed8080412

**Published:** 2023-08-13

**Authors:** Kylychbek Istamov, Mher Beglaryan, Olga Goncharova, Konushbek Sakmamatov, Bolot Kyrbashov, Mukadas Mamytova, Indira Zairova, Gulzat Alumkylova, Divya Nair

**Affiliations:** 1School of Medicine, Osh State University, Osh City 723500, Kyrgyzstan; mamytova28@gmail.com (M.M.); zit-0502@mail.ru (I.Z.); 2Tuberculosis Research and Prevention Center, Yerevan 0014, Armenia; works@mher.pro; 3National Center for Phthisiology, Bishkek 720000, Kyrgyzstan; goncharova.ncph@gmail.com (O.G.); bolotkyrbashov@gmail.com (B.K.); gulzatalumkulova1986@gmail.com (G.A.); 4Faculty of Medicine, Ala-Too International University, Bishkek 720000, Kyrgyzstan; ksakmamatov@gmail.com; 5International Union Against TB and Lung Disease (The Union), 75001 Paris, France; divya.nair@theunion.org

**Keywords:** transients and migrants, Kyrgyzstan, Central Asia, treatment outcomes, vulnerable populations, key population, patient delay, diagnostic delay, treatment delay, SORT-IT

## Abstract

Migrants are at increased risk of developing tuberculosis (TB) and have poor treatment outcomes. The National TB program (NTP) of the Kyrgyz Republic recognizes two types of migrants: internal (intra-country) and external (inter-country) migrants. This cohort study compared the characteristics, timeliness of diagnosis and treatment initiation, and treatment outcomes of TB patients (internal migrant vs. external migrant vs. non-migrant) identified during treatment in the country in 2021. The TB treatment register and treatment cards of 5114 patients (156 internal, 430 external, and 4528 non-migrants) were reviewed. Risk factors (unemployment, smoking, alcohol use, and homelessness) were higher (*p*-value < 0.001) in internal (84%) than in external migrants (66%) and non-migrants (43%). The median delay in seeking care post-symptom onset was longer (*p*-value= 0.03) in external (30 days) than in internal migrants (21 days) and non-migrants (25 days). Successful treatment outcomes for drug-sensitive TB were higher in internal (89%, *p*-value = 0.012) and external migrants (86%, *p*-value = 0.001) than in non-migrants (78%). Internal and external migrants should be separately considered with respect to TB care and monitoring under the NTP. Success rates seem to be high in migrants, but our findings may be biased, as migrants with poor healthcare access may remain undetected and untreated and have undocumented poor outcomes.

## 1. Introduction

The World Health Organization’s (WHO) End Tuberculosis (TB) strategy aims to reduce the incidence of and mortality due to TB by 80% and 90% by 2030, respectively, compared to 2015 levels. Early detection and timely treatment of patients with TB is the cornerstone of this strategy. The strategy also recommends focusing on populations that are highly vulnerable to infection and have poor access to health care [1].

Migrants are one such vulnerable group, as they are predisposed to acquiring TB infection due to poor living conditions, limited access to healthcare, and, consequently, pre-existing health issues. They are also more vulnerable to delays occurring during the process of diagnosis and treatment due to a lack of access to health care and less protection against stigma and marginalization. This issue, in turn, increases the probability of TB transmission, severe forms of disease at presentation, and unsuccessful treatment outcomes [2]. Therefore, migrants have been recognized as a “key population” requiring special attention in TB prevention and control efforts [3]. However, achieving this goal remains a challenge, especially in low- and middle-income countries (LMICs) [4,5].

The Kyrgyz Republic is an LMIC in Central Asia that has not been able to make significant progress in the achievement of End TB goals. The incidence of TB has fluctuated over the years and is currently 130 per 100,000 people. The treatment success rate has remained stagnant at around 80% among new cases and 70% among retreatment cases, respectively, since 2015. The Kyrgyz Republic is among the 30 countries with the highest burden of multi-drug resistant (MDR) TB [6].

The Kyrgyz economy is heavily reliant on labor migration [7]. The migrant population includes individuals who migrate within the country (internal migrants) and from neighboring countries (external migrants). A substantial proportion of diagnosed TB patients (45%) in the capital city of Bishkek were migrants in the period 2012–2013 [8].

Successful TB treatment outcomes among migrants ranged from 71% during the period 2012–2013 in Bishkek city to 74% during the period 2015–2017 in the Chui region, which were lower than the national average (82%), and the WHO recommended a target of 85% at that time [8,9,10]. Most of the migrants with TB are Kyrgyz citizens and, therefore, are expected to have access to health care comparable to non-migrants. The reasons for sub-optimal success rates among these populations are not well understood and could be attributed to a lack of awareness and treatment interruptions due to their highly mobile lifestyle [11]. These studies considered migrants to be a single group, and they did not differentiate between internal and external migrants.

Since the above-mentioned studies were published, there have been changes that could have influenced the TB situation among migrants in the Kyrgyz Republic. Firstly, the National TB Program (NTP) has taken steps to facilitate timely diagnosis and treatment of TB. These include community-based information, education and communication campaigns, adherence support systems, improvement in diagnostic infrastructure, tracing of patients who are lost to follow-up, the digitization of health records, and short course therapy for MDR TB. Secondly, the COVID-19 pandemic has impacted the scale of migration due to the restrictions placed on population mobility and uncertainties around employment [12].

In view of these changes, there is a need to reassess the current situation of TB among migrants in the country. Additionally, comparisons between the delays in treatment initiation and treatment outcomes between internal, external, and non-migrants can help identify the steps in the care cascade at which the migrants may require special attention. This assessment can provide key insights to identify and quantify gaps in the diagnosis and care of migrants with TB and could help to identify priority areas within the two groups of migrants in programmatic settings.

Therefore, a study was undertaken to compare the clinical characteristics, timeliness of diagnosis and treatment initiation, and treatment outcomes between internal, external, and non-migrant populations for whom treatment was initiated in the Kyrgyz Republic in 2021.

## 2. Materials and Methods

### 2.1. Study Design

A cohort study involving analysis of routinely collected programmatic data.

### 2.2. Study Setting

#### 2.2.1. General Setting

The Kyrgyz Republic is a Central Asian country that borders Kazakhstan, Tajikistan, Uzbekistan, and China. It has a population of approximately seven million people [13]. The country is divided into seven regions (oblasts); each region is further divided into districts (rayons). The cities of Bishkek and Osh are to be considered independent cities and do not belong to any region. Public health services in the country are delivered through a three-tier public healthcare system. Each Kyrgyz national is issued a Personal Identification Number (PIN), which is considered to be valid proof of identity and address and is required for registration with the public health care system.

#### 2.2.2. Specific Setting

The NTP delivers TB prevention and control services through a network of dedicated TB diagnostic and care facilities, which is in line with WHO guidelines. All TB care services are provided free of cost to Kyrgyz nationals. Foreign nationals who have invested in the government-sponsored insurance scheme, which is known as the Mandatory Health Insurance Fund, are entitled to a 50% discount on their healthcare costs [14]. Those uninsured foreign nationals are registered for treatment under the NTP if they are able to make out-of-pocket payments to cover treatment costs.

Diagnosis of tuberculosis: Individuals with symptoms can avail medical care at any health facility in the country. In the public health system, the first point of contact for patients is usually a polyclinic, which serve as the primary health centers in the country. If the individual is considered to have presumptive TB, sputum samples are collected for smear microscopy and Xpert MTB/RIF testing. If tubercular mycobacteria are detected via any of these tests, the individual is diagnosed with TB. Sputum smear microscopy is available in all polyclinics in the country, while the Xpert MTB/RIF test is available at a few designated laboratories in each region. A specimen collection and transport system is operational in each region for the transport of samples from polyclinics to the linked Xpert MTB/RIF laboratories. A diagnosis of TB can also be made solely on the basis of clinical assessment by a physician when there is a strong clinical suspicion, even in the absence of bacteriological confirmation.

Initiation of treatment: Once a patient is confirmed as having TB, he or she is referred to a TB physician. Treatment is initiated as soon as possible, while drug sensitivity results are awaited. At the time of treatment initiation, the TB physician initiates a TB treatment card for the patient and assigns a unique registration number to the patient. The details of the patient are also entered into a TB treatment register, which is maintained at the treatment facility.

If resistance (poly resistance, rifampicin resistance, or MDR or extensive drug resistance (XDR)) is detected in any of the drug sensitivity tests, the treatment regimen is changed by the TB physician in consultation with other physicians at the regional level.

The treatment card and register are maintained and updated by TB physicians at the treating facility throughout the duration of treatment until the treatment outcome is assessed. Since 2021, efforts have been made to digitize the treatment card on the National Electronic TB Register.

TB among migrants in the Kyrgyz Republic:

The NTP considers the following issues as risk factors for TB: a history of contact with TB, homelessness, drug use, smoking, alcohol use, a history of imprisonment, working in the field of TB care, working in health care, unemployment, comorbidities, and migration (external and internal).

Decisions on external migrant or internal migrant status were based on the area of residence and the travel history of the patient (Box 1).

Box 1Criteria for designating external and internal migrant status according to the National Tuberculosis Program of the Kyrgyz Republic.
External Migrants:Kyrgyz nationals:
○If the place of residence (based on Personal Identification Number (PIN) or self-reported) belongs to the same region/oblast as the detecting facility, AND;○If the patient has traveled outside of the Kyrgyz Republic in the preceding three months and resided in the visited location for more than three months.
Foreign Nationals: A foreign national who is diagnosed as having TB during his/her stay in the Kyrgyz RepublicInternal Migrants:If the place of residence is located in a different region/oblast than the detecting facility, AND;If the patient has been staying in the region for more than six months.


### 2.3. Study Population

The study included all individuals with TB for whom treatment was initiated in the Kyrgyz Republic in 2021.

### 2.4. Data Sources, Collection, and Entry

Data for study variables were extracted from the NTP’s electronic TB register and cross verified with paper-based TB treatment registers and cards. Data were entered into an MS Excel 2016 (Microsoft Corporation, Redmond, WA, USA, 2016) spreadsheet.

### 2.5. Operational Definitions

Migrants: Any patient for whom the risk factor “Internal migrant” or “External migrant” was entered into the treatment register were considered to be migrants, irrespective of whether they had any other concomitant risk factors.Successful treatment outcomes: If the treatment outcomes in the treatment were reported as “Cured” or “Treatment” and completed in the treatment register.Unsuccessful outcomes: If any of these outcomes is reported in the treatment register— “Treatment failure”, “Death”, “Loss to follow up”, or “Not evaluated”—treatment was deemed unsuccessful. Also, those patients for whom no treatment outcome was mentioned were considered not to have been evaluated and were, hence, categorized under “Unsuccessful outcomes”.Calculation of delays: Patient delay was the time between the onset of symptoms and the first visit to a health facility. Diagnostic delay was the time between first visit to a health facility and the confirmation of TB. Treatment delay was the time between diagnosis of TB and treatment initiation. Health system delay comprised of diagnostic delay and treatment delay [15].

### 2.6. Statistical Analysis

Data was analyzed using STATA^®^ (version 16.0 Copyright 1985–2019 StataCorp LLC, Lakeway, TX, USA). Categorical variables (gender, population risk group, type of TB etc.) were summarized as frequencies and percentages. Delays in treatment and diagnosis were summarized as medians with inter-quartile range, and statistical comparisons were performed using the Kruskal–Wallis test. A chi-square test was used to compare socio-demographic and clinical characteristics, as well as treatment outcomes between internal migrants, external migrants, and non-migrants. A *p*-value < 0.05 was considered to be statistically significant.

## 3. Results

During 2021, 5114 individuals were initiated on anti-TB treatment in the Kyrgyz Republic. Of these, 156 (3%) were internal and 430 (8%) w ere external migrants.

### 3.1. Sociodemographic Characteristics of Migrants and Non-Migrants

The socio-demographic and clinical characteristics of the study population are shown in Table 1. Higher proportions of internal migrants (129, 83%) and external migrants (398, 93%) belonged to the age group 15–59 years than non-migrants (3191, 71%). The external migrants were mostly registered in the Osh (129, 30%), Jalal-Abad (83, 19%), and Chui (66, 15%) regions. Chui region and Bishkek city had the highest number of registered internal migrants (135, 87%). All but ten identified external migrants were Kyrgyz citizens (420, 98%).

Higher proportions (*p* < 0.001) of internal (134, 86%) and external migrants (283, 66%) had co-existing risk factors than non-migrants (1904, 42%). Unemployment was the most common risk factor in each of the groups, though it was highest in internal migrants (114, 73%), followed by external migrants (246, 57%) and non-migrants (1371, 30%).

### 3.2. Clinical Characteristics of Migrants and Non-Migrants

Clinical characteristics of migrants and non-migrants are shown in Table 2. Treatment was initiated at a facility in the same region as the detecting facility for all of the patients. Newly diagnosed TB was higher among internal (118, 76%) and external migrants (355, 83%) than non-migrants (3164, 70%). MDR was higher among internal migrants (33, 21%) than the other groups (10–13%).

For about 20% of the internal and external migrants, HIV status was not known due to their refusal to be tested, whereas it was 35% among non-migrants. Data regarding the history of previous TB treatment, the site of TB treatment, and drug resistance were not available in 8% of the non-migrant individuals.

### 3.3. Patient, Diagnostic, and Treatment Delays for Migrants and Non-Migrants

Valid data for the calculation of the time between the onset of symptoms and the first contact with a heath care provider were available for 86–91% patients across the three groups (internal migrants, external migrants, and non-migrants). Valid data for the time between first contact with health care provider and the diagnosis of TB were available for about 65% of internal and external migrants and 60% of non-migrants. Valid data for the time between the diagnosis of TB and the initiation of anti-TB treatment were available for 54% of internal migrants, 42% of external migrants, and 46% of non-migrants.

The median (IQR) time taken to seek treatment after the onset of symptoms (patient delay) was longer (*p* = 0.03) in external migrants [30 (14–59) days] than in internal migrants [21 (9–56) days] and non-migrants [25 (11–49) days]. The health system delay was similar across the three groups, with the median delay ranging between 6 and 8 days (Table 3).

### 3.4. Treatment Outcomes of Migrants and Non-Migrants

Treatment outcomes among internal, external, and non-migrants stratified by type of drug resistance are shown in Figure 1. Among drug-sensitive TB patients, the successful treatment outcomes in internal migrants (86, 89%) and external migrants (259, 86%) were similar (pairwise Chi square test, *p* = 0.510). Successful treatment outcomes among drug-sensitive TB patients were relatively better in internal migrants than non-migrants (pairwise Chi square test, *p* = 0.012). Similarly, successful treatment outcomes among drug-sensitive TB patients were relatively better (pairwise Chi square test, with a *p* of 0.001 for external migrants than non-migrants (2539, 78%). Among drug-resistant TB patients, the successful outcomes were similar (*p* = 0.922) across internal migrants (43, 73%), external migrants (90, 73%), and non-migrants (643, 71%).

## 4. Discussion

This investigation was the first nationwide study that explored the characteristics, treatment outcomes, and timeliness of diagnosis and initiation of treatment for TB among internal migrants, external migrants, and non-migrants in the Kyrgyz Republic. Almost all risk factors (unemployment, smoking, alcohol use, and homelessness) were higher in internal migrants than in external migrants and non-migrants. Similarly, MDR was higher among internal migrants. External migrants had a slightly longer delay in contacting a health care provider for their symptoms. Both external and internal migrants had higher success rates in drug-sensitive TB than non-migrants.

The study has certain strengths. Firstly, this study addressed a national research priority and adds to the limited evidence about TB and its outcomes among a vulnerable population group in the Central Asian region. Secondly, the study used country-wide data for one year, which limited the potential for selection bias. Thirdly, this study looked at internal and external migrants separately. This method brought to light certain differences between the two groups, which could not have been identified if they had been analyzed together as migrants. Finally, the conductance and reporting of the study was in accordance with the STROBE (Strengthening the Reporting of Observational Studies in Epidemiology) guidelines [16].

The study had certain limitations. Firstly, the study followed the NTP’s classifications of internal and external migrants, which could be different from those followed in other countries. In fact, there is a lack of global consensus on the definition of migrants [17]. Secondly, the dates required for the calculation of patient, diagnostic, or treatment delays were missing or invalid in about 40% of the patients; therefore, the estimates of patient and health system delays may not reflect the true picture. Thirdly, some of the cases with missing outcomes may have been under ongoing treatment, particularly drug-resistant cases, given their prolonged treatment. However, the current study treats all missing outcomes as unsuccessful because the data provided no means of identifying ongoing cases of treatment. Consequently, our results can be biased toward underestimating the success rate. This bias may be more pronounced among drug-resistant cases. Fourthly, the NTP’s electronic TB register contains only those patients for whom treatment was initiated; therefore, it was not possible to assess the number of presumptive TB patients and pre-treatment loss to follow-up among the different population groups. Fifthly, the NTP’s electronic TB register lacks data regarding treatment adherence, hence it was not possible to explore if the differences in outcomes were influenced by treatment adherence. Sixthly, a qualitative exploration of the reasons for and patterns of migration among internal and external migrants, their health seeking behavior, and their perceptions regarding TB may have helped us to better understand some of the differences in the two groups. However, this action was not within the scope of the study.

The study’s findings have important programmatic implications. Clearly, there is a difference between internal and external migrants. A larger proportion of internal migrants (86%) have co-existing risk factors than external migrants and non-migrants. Unemployment rates (73%) were much higher in internal migrants. MDR was higher among internal migrants. Internal migrants are, therefore, likely to be more disadvantaged than external migrants and non-migrants. In this context, it is pertinent for the NTP to pay special attention to the health care needs of the internal migrants and consider them as a separate group for the purposes of monitoring and evaluation.

Patient delay, that is, delay in seeking care after the onset of symptoms, was higher in external migrants. There are no studies from this region that have quantified delays in care seeking, diagnosis, and treatment initiation among migrants. Studies from Tajikistan and Uzbekistan have documented care-seeking delays of 21–27 days among the general population [18,19]. In the Kyrgyz Republic context, external migrants are mostly Kyrgyz Republic nationals who have returned after staying/working abroad for some time. The NTP’s classification of external migrants, therefore, cannot be equated to the traditional use of “external migrants”, which refers to refugees and displaced populations [20]. The delay in care seeking, as seen in this study, could be due to issues faced by external migrants in procuring the necessary documentation required for registering to receive care within the public health system and a lack of access to health care services upon their return to the Kyrgyz Republic [21]. Lack of legal registration has been reported as a significant barrier to accessing health services among external migrants with TB in the neighboring country of Kazakhstan [5]. This issue is an area that merits further exploration using qualitative techniques.

Interestingly, both external and internal migrants had higher treatment success rates (86% and 89%, respectively) among patients with drug-sensitive TB, while non-migrants had a success rate of 78%. In drug-resistant TB, success rates were similar across all groups (71–73%). This figure is an improvement on the success rates of 61–74% reported among migrants in different regions of the Kyrgyz Republic [8,9,22]. This observation perhaps means that the NTP pays special attention to the treatment and follow-up of patients with risk factors. Thus, as migrants have a higher prevalence of risk factors, they would have been accorded priority.

However, it is prudent not to become complacent as a result of these findings. It is important to consider the highly mobile nature of the migrant population and the possibility that individuals are not able to dedicate time to diagnosis and treatment of TB [11]. These individuals could be the missed (undetected or detected but not treated) patients, who might actually have more severe forms of disease and poorer outcomes. Migrants for whom treatment has been initiated (study population) could be those who had better access to health care and more stable lifestyles. Therefore, the current study of treatment outcomes may be influenced by survivorship bias, which may be more pronounced in a vulnerable population, such as migrants. Currently, the NTP of the Kyrgyz Republic relies on passive case-finding approaches for TB case detection. Initiatives like targeted active case finding, which have been shown to improve detection of TB in at risk population groups, can be adopted [23,24]. Such initiatives can be started as pilot projects, which can provide opportunities to systematically estimate the actual burden of TB in these groups and their treatment outcomes. The NTP should also put in place a system of registering patients at the time that they present with symptoms (presumptive TB patients) to ensure that they can be tracked and pre-treatment loss to follow-up can be assessed and addressed. It is also worthwhile to explore the reasons for low treatment success rates in non-migrants.

The study also found that data were missing with regard to important variables, like drug resistance patterns and treatment outcomes. There were also instances of invalid data, especially in the recording of dates. This issue could be attributed to the errors occurring during the manual transcription of dates in the paper-based treatment registers and cards, which were in use during the study period. Moving forward, it is expected that the recently introduced electronic treatment cards with in-built quality checks would help to mitigate data quality issues, like implausible values and incompleteness. Data quality should be monitored as part of routine program monitoring and evaluation. The NTP should also educate the program’s staff regarding the importance of maintaining high data quality standards.

## 5. Conclusions

In the Kyrgyz Republic, internal migrants have higher levels of risk factors (unemployment, smoking, alcohol use, and homelessness) than external migrants and non-migrants. External migrants take more time to seek care after the onset of symptoms. In spite of this fact, migrants have better treatment outcomes, possibly due to the NTP’s specialized attention toward this vulnerable group and survivorship bias. Moving forward, the NTP needs to consolidate these gains to ensure successful treatment outcomes in migrant populations, as well as try to ensure the same outcomes in non-migrants. Also, it is imperative to ensure that no migrant with TB remains undetected and untreated due to issues with health facility access and reliance on passive approaches used in TB case finding.

## Figures and Tables

**Figure 1 tropicalmed-08-00412-f001:**
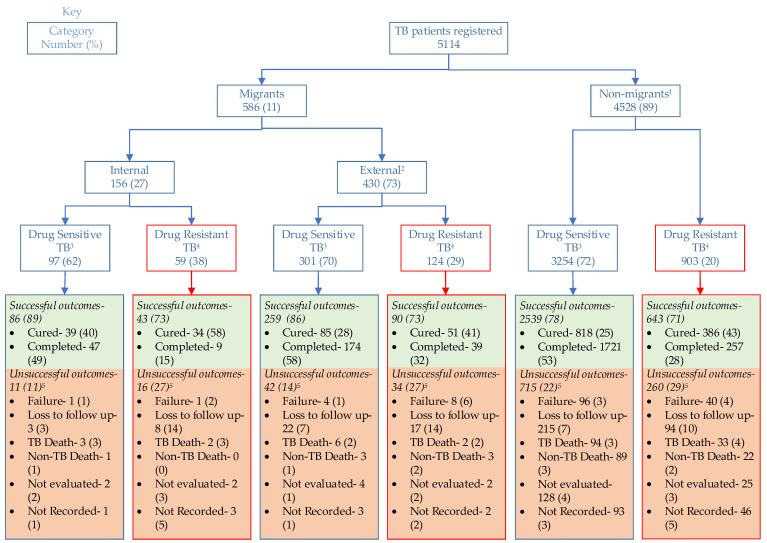
Treatment outcomes of migrants and non-migrants for whom anti-tuberculosis treatment was initiated in the Kyrgyz Republic during 2021. ^1^ Data on type of resistance were not available in 371 non-migrants. ^2^ Data on type of resistance were not available in five external migrants. ^3^ Among drug-sensitive TB patients, Chi square tests of independence between three migrant categories (internal migrants, external migrants, and non-migrants) and binary treatment outcomes (successful or unsuccessful) yielded a *p*-value of *p* < 0.001. ^4^ Includes poly resistant, rifampicin resistant, multi-drug resistant, and extensively drug-resistant forms. Among drug-resistant TB patients, Chi square tests of independence between three migrant categories (internal migrants, external migrants, and non-migrants) and binary treatment outcomes (successful or unsuccessful) yielded a *p*-value of *p* = 0.922. ^5^ TB Deaths are deaths attributable to TB, and Non-TB Deaths are deaths that occurred due to other reasons (as recorded in the electronic TB registers).

**Table 1 tropicalmed-08-00412-t001:** Socio-demographic characteristics of migrants and non-migrant individuals for whom anti-tuberculosis treatment was initiated in the Kyrgyz Republic during 2021.

Characteristic	Total (N = 5114)	Internal Migrants (N = 156)		External Migrants (N = 430)		Non-Migrants (N = 4528)		*p*-Value
		*n*	(%)	*n*	(%)	*n*	(%)	
Age (in years)								<0.001
0–14	276	5	(3.2)	13	(3.0)	258	(5.7)	
15–29	1459	54	(34.6)	219	(50.9)	1186	(26.2)	
30–44	1160	49	(31.4)	113	(26.3)	998	(22.0)	
45–59	1099	26	(16.7)	66	(15.4)	1007	(22.2)	
60 and above	1105	22	(14.1)	19	(4.4)	1064	(23.5)	
Not recorded	15	0	(0)	0	(0)	15	(0.3)	
Gender								0.061
Male	2933	85	(54.5)	269	(62.6)	2579	(56.9)	
Female	2181	71	(45.5)	161	(37.4)	1949	(43.0)	
Region of registration								<0.001
Chui	1083	92	(58.9)	66	(15.4)	925	(20.4)	
Jalal-Abad	971	4	(2.6)	83	(19.3)	884	(19.5)	
Osh region	954	2	(1.3)	129	(30.0)	823	(18.2)	
Bishkek city	921	43	(27.6)	33	(7.7)	845	(18.6)	
Osh city	305	0	(0)	29	(6.7)	276	(6.1)	
Batken	305	3	(1.9)	44	(10.2)	258	(5.7)	
Issykul	222	2	(1.3)	29	(6.7)	191	(4.2)	
Naryn	171	6	(3.8)	6	(1.4)	159	(3.5)	
Talas	180	4	(2.6)	11	(2.6)	165	(3.6)	
Not recorded	2	0	(0)	0	(0)	2	(<1)	
Citizenship								
The Kyrgyz Republic	5104	156	(100)	420	(98)	4528	(100)	
Other countries	10	0	(0)	10	(2)	0	(0)	
Risk factors ^1^								
At least one risk factor	2321	134	(85.9)	283	(65.8)	1904	(42.1)	<0.001
Unemployed	1731	114	(73.1)	246	(57.2)	1371	(30.3)	<0.001
Smokers	383	32	(20.5)	33	(7.7)	318	(7.0)	<0.001
Alcohol users	194	18	(11.5)	12	(2.8)	164	(3.6)	<0.001
Homeless	64	11	(7.1)	5	(1.2)	48	(1.1)	<0.001
Ex-prisoners	56	7	(4.5)	2	(0.5)	47	(1.0)	<0.001
Health care workers	21	1	(0.6)	2	(0.5)	18	(0.4)	0.881
Intravenous drug users	13	0	(0.0)	2	(0.5)	11	(0.2)	0.556

TB: tuberculosis. ^1^ A patient could have >1 risk factors. For each risk factor, the comparisons were made between all individuals who had a particular risk factor and all individuals who did not have that particular risk factor.

**Table 2 tropicalmed-08-00412-t002:** Comparison between diagnostic and clinical characteristics of migrants and non-migrant individuals for whom anti-tuberculosis treatment was initiated in the Kyrgyz Republic during 2021.

Characteristic	Total (N = 5114)	Internal Migrants (N = 156)		External Migrants (N = 430)		Non-Migrants (N = 4528)		*p*-Value
		*n*	(%)	*n*	(%)	*n*	(%)	
Type of facility at which diagnosis was established								<0.001
Polyclinic in same region ^1^	4114	132	(84.6)	400	(93.0)	3582	(79.1)	
Specialised TB facility in same region ^1^	996	24	(15.4)	30	(6.9)	942	(20.8)	
Not recorded	4	0	(0)	0	(0)	4	(0.1)	
History of previous TB treatment								<0.001
New	3637	118	(75.6)	355	(82.5)	3164	(69.8)	
Previously treated	1080	36	(23.1)	68	(15.8)	976	(21.5)	
Not recorded	397	2	(1.3)	7	(1.6)	388	(8.6)	
Site of TB								<0.001
Pulmonary	3831	131	(83.9)	360	(83.7)	3340	(73.8)	
Extrapulmonary	902	25	(16.0)	65	(15.1)	812	(17.9)	
Not recorded	381	0	(0.0)	5	(1.2)	376	(8.3)	
Drug resistance								<0.001
Sensitive	3652	97	(62.1)	301	(70.0)	3254	(71.8)	
Poly resistant	262	17	(10.9)	29	(6.7)	216	(4.8)	
Rifampicin resistant	242	8	(5.1)	36	(8.4)	198	(4.4)	
Multi drug resistant	530	33	(21.1)	55	(12.8)	442	(9.8)	
Extensively drug resistant	52	1	(0.6)	4	(0.9)	47	(1.0)	
Not recorded	376	0	(0)	5	(1.2)	371	(8.2)	
HIV co-infection								<0.001
Present	85	5	(3.2)	9	(2.1)	78	(1.5)	
Absent	3320	123	(78.9)	333	(77.4)	3288	(63.4)	
Refused to be tested	1709	28	(17.9)	88	(20.5)	1821	(35.1)	

TB: tuberculosis; HIV—Human immunodeficiency virus. ^1^ Same region—facility at which diagnosis was made was located in the same region as the facility at which treatment was initiated.

**Table 3 tropicalmed-08-00412-t003:** Comparison between patient, diagnostic, and treatment delays in migrant and non-migrant individuals for whom anti-tuberculosis treatment was initiated in the Kyrgyz Republic during 2021.

Stages	Internal Migrants (N=156)				External Migrants (N=430)				Non-Migrants (N=4528)			*p*-Value
	n (%) ^1^	Median Days	(IQR)		n (%) ^1^	Median Days	(IQR)		n (%) ^1^	Median Days	(IQR)	
Onset of symptoms to first contact with heath care provider (Patient delay)	134 (86)	21	(9-56)		391 (91)	30	(14-59)		3985 (88)	25	(11-49)	0.003
First contact with heath care provider to diagnosis of TB (Diagnostic delay)	103 (66)	6	(3-14)		275 (64)	4	(2-18)		2712 (60)	5	(2-14)	0.5
Diagnosis of TB to initiation of treatment (Treatment delay)	84 (54)	5	(2-18)		199 (46)	4	(1-13)		1924 (42)	4	(2-14)	0.5

Health system delay ^2^	145 (93)	8	(3-20)		379 (88)	6	(2-20)		3914 (86)	7	(3-20)	0.28

IQR—inter quartile range in days. ^1^ Number and percentage of total patients in each category for whom valid dates were available to calculate the delays at each stage. ^2^ Health system delay—Diagnostic delay and Treatment delay.

## Data Availability

Requests to access these data should be sent to the corresponding author.

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
