# Peer review of "Delays in Treatment Initiation and Treatment Outcomes in Patients with Tuberculosis in the Kyrgyz Republic: Are There Differences between Migrants and Non-Migrants?"

_tropicalmed, 2023, doi:10.3390/tropicalmed8080412_

Round 1
Reviewer 1 Report
Delays in treatment initiation and treatment outcomes in patients with tuberculosis in the Kyrgyz Republic: Are there differences between migrants and non-migrants?
Kylychbek Istamov et al .
Tropical Medicine and Infectious Disease – 2512182-peer review-v1.pdf
The experience of migration is a key determinant of health and well-being. Migrants are among the most vulnerable members of society and an essential part of the End TB Strategy. The authors rightly aim to evaluate the situation of the migrant population in the Kyrgyz Republic with respect to tuberculosis diagnosis, treatment initiation, and outcome. Istamov and collaborators employ the National TB program data to compare sociodemographic and clinical characteristics of migrant and non-migrant populations in the Kyrgyz Republic. Importantly they distinguish between internal and external migrants, which adds to their study. This is a sound epidemiological study of the state of tuberculosis in the Kyrgyz Republic and of specific interest to this country, which can help inform policy decision makers to improve care; it is unclear how the knowledge gained in the study can be translated to other countries around the world. The study should be published with some modifications.
General comments –
1. As the authors readily accept, there is an inclusion bias.
a. The authors could strengthen the study by adding "deaths attributable" to tuberculosis in the National Health Data Bases for all three groups, if available.
2. The authors claim that "since the study included country wide data of all patients who were initiated on treatment in 2021, tests for statistical significance were not carried out." (lines 167-169). Not clear why this is the case. Statistical significance tests should be performed.
a. In particular, the statistical significance of the different risk factors need to be performed.
3. Dividing the treatment delay measurement into three periods is very important and helps understand the country's different aspects of the Tuberculosis attention program.
4. The authors do not mention if there is a measurement for adherence to treatment, which is a critical element for the health outcome in TB treatment.
5. There are several questions that remain that might be important to follow:
a. Internal migrants have a higher proportion of MDR – TB
b. How can the survival vias be eliminated?
Specific points
1. Line 149 – "Migrants: any patient against whom …" against should not be used in this context.
2. The diagnostic criteria are not always what is recommended by WHO. Specifically: "Diagnosis of TB can also be made solely based on clinical assessment of the physician, especially in extrapulmonary cases." Line 116 -117. These cases may mistake other mycobacteria for M tuberculosis and should not be included , or followed independently. The authors do state that bacilloscopic analysis and Xpert testing are used in some cases.
3. The study would benefit from comparison to studies of tuberculosis in migrant populations in other countries, mainly due to their results that show a better outcome in migrant populations. A rapid not-comprehensive search found these articles, which may enrich this study's discussion.
· Pareek, M., Greenaway, C., Noori, T. et al. The impact of migration on tuberculosis epidemiology and control in high-income countries: a review. BMC Med 14, 48 (2016).
· Silva DR, Mello FCQ, Johansen FDC, Centis R, D'Ambrosio L, Migliori GB. Migration and medical screening for tuberculosis. J Bras Pneumol. 2023 Apr 28;49(2):e20230051. doi: 10.36416/1806-3756/e20230051. PMID: 37132706; PMCID: PMC10171264.
· Meaza A, Tola HH, Eshetu K, Mindaye T, Medhin G, Gumi B. Tuberculosis among refugees and migrant populations: Systematic review. PLoS One. 2022 Jun 9;17(6):e0268696. doi: 10.1371/journal.pone.0268696. PMID: 35679258; PMCID: PMC9182295.
· Ejeta E, Beyene G, Balay G, Bonsa Z, Abebe G. Factors associated with unsuccessful treatment outcome in tuberculosis patients among refugees and their surrounding communities in Gambella Regional State, Ethiopia. PloS one. 2018;13(10):e0205468. doi: 10.1371/journal.pone.0205468
· Lomtadze N, Aspindzelashvili R, Janjgava M, Mirtskhulava V, Wright A, Blumberg H, et al.. Prevalence and risk factors for multidrug-resistant tuberculosis in the Republic of Georgia: a population-based study. 2009;13(1):68–73.

Reviewer 2 Report
In the results, it would be convenient to provide the percentage of previous resistance to anti-TB drugs, to show an increasing or decreasing comparison with the data presented for 2021.
For the publication, it is necessary in Conclusions, to perform a better analysis of the current failures in the TB program among non-immigrant patients, since it is the program that should show better indicators and it is not so: therapeutic success rates are low, ignorance of the type of TB in 8% (376 patients) of cases, the loss of treatment follow-up and the higher percentage of patients with a risk factor for TB, says that the diagnosis is not being actively sought, which forces us to conclude that the program must improve and say in what way it should be programmed to improve.
It should also be mentioned as a barrier for immigrants to have timely medical attention, diagnosis and treatment, not only the delay in enrollment in the national health system, but the collection of 50% of the costs, since it must be difficult to cover them in the immigrant situation.
Reviewer 3 Report
The authors present findings on their analysis of the effects of delays in treatment initiation and treatment outcomes in patients with tuberculosis in the Kyrgyz Republic to see if there differences between migrants and non-migrants. The premise is that migrants would be in poorer health and have less access to diagnosis and treatment than non-migrants and therefore would fair poorer outcomes.
The study is well designed and the data is presented in a clear and concise form. The manuscript is generally well written and the only concern I have is the discussion of results and the abstract.
For example, the authors write in the abstract that “While success rates seem high in migrants, further investigation is required to assess 28 if there are undetected/ untreated patients among them resulting in survivorship bias.”
One in fact would not expect undetected/untreated patients to affect the results since they would not be included in the delays (patient, diagnosis, or treatment delays) or in the treatment outcomes. Speculation as to what factors could cause the bias in survivorship or outcome success in general need to also be expanded in the conclusion section. It is very important that the authors suggest specific areas to investigate in order to identify the reason(s) for this bias. For example, the large difference in sample size between the three groups. Internal migrants made up only 3% of the study subjects and external migrants only 8% of the study subjects. This in my opinion is the only weakness of the study.
Round 2
Reviewer 1 Report
After reviewing the response of the author, I think the authors have addressed all the relevant concerns and the document is ready for publication.
Author Response
Reviewer's comment: After reviewing the response of the author, I think the authors have addressed all the relevant concerns and the document is ready for publication.
Response: We thank the reviewer for his/her comments and inputs which have helped in improving the manuscript.